# Targeted MRM Quantification of Urinary Proteins in Chronic Kidney Disease Caused by Glomerulopathies

**DOI:** 10.3390/molecules28083323

**Published:** 2023-04-09

**Authors:** Alexey S. Kononikhin, Alexander G. Brzhozovskiy, Anna E. Bugrova, Natalia V. Chebotareva, Natalia V. Zakharova, Savva Semenov, Anatoliy Vinogradov, Maria I. Indeykina, Sergey Moiseev, Irina M. Larina, Evgeny N. Nikolaev

**Affiliations:** 1Center for Molecular and Cellular Biology, Skolkovo Institute of Science and Technology, Bolshoy Boulevard 30, Bld. 1, 121205 Moscow, Russia; 2Kulakov National Medical Research Center for Obstetrics, Gynecology and Perinatology of the Ministry of Health, 117997 Moscow, Russia; 3Emanuel Institute for Biochemical Physics, Russian Academy of Science, Kosygina Str. 4, 119334 Moscow, Russia; 4Nephrology Department, Sechenov First Moscow State Medical University, Trubezkaya 8, 119048 Moscow, Russia; 5Department of Internal Medicine, Lomonosov Moscow State University, GSP-1, Leninskie Gory, 119991 Moscow, Russia; 6Moscow Institute of Physics and Technology, 141700 Dolgoprudny, Russia; 7Institute of Biomedical Problems, Russian Federation State Scientific Research Center, Russian Academy of Sciences, Khoroshevskoe Shosse 76A, 123007 Moscow, Russia

**Keywords:** urine proteomics, multiple-reaction monitoring, mass spectrometry, proteomics, chronic kidney disease

## Abstract

Glomerulopathies with nephrotic syndrome that are resistant to therapy often progress to end-stage chronic kidney disease (CKD) and require timely and accurate diagnosis. Targeted quantitative urine proteome analysis by mass spectrometry (MS) with multiple-reaction monitoring (MRM) is a promising tool for early CKD diagnostics that could replace the invasive biopsy procedure. However, there are few studies regarding the development of highly multiplexed MRM assays for urine proteome analysis, and the two MRM assays for urine proteomics described so far demonstrate very low consistency. Thus, the further development of targeted urine proteome assays for CKD is actual task. Herein, a BAK270 MRM assay previously validated for blood plasma protein analysis was adapted for urine-targeted proteomics. Because proteinuria associated with renal impairment is usually associated with an increased diversity of plasma proteins being present in urine, the use of this panel was appropriate. Another advantage of the BAK270 MRM assay is that it includes 35 potential CKD markers described previously. Targeted LC-MRM MS analysis was performed for 69 urine samples from 46 CKD patients and 23 healthy controls, revealing 138 proteins that were found in ≥2/3 of the samples from at least one of the groups. The results obtained confirm 31 previously proposed CKD markers. Combination of MRM analysis with machine learning for data processing was performed. As a result, a highly accurate classifier was developed (AUC = 0.99) that enables distinguishing between mild and severe glomerulopathies based on the assessment of only three urine proteins (GPX3, PLMN, and A1AT or SHBG).

## 1. Introduction

Chronic kidney disease (CKD) is a non-specific condition with clinical symptoms that can be caused by various reasons [1]. This socially significant pathology affects up to 13.4% of people [2] and carries a high risk of disability, requiring high-cost treatments such as hemodialysis, peritoneal dialysis, and kidney transplants [3,4]. At the same time, the severity of CKD manifestations does not always reflect the degree of renal damage [5] and may largely depend on the nature of the underlying nephropathy. In recent years, there has been a steady increase in the incidence of severe glomerular diseases—especially focal segmental glomerulosclerosis (FSGS), which has increased by 41% over the past 10 years, approaching diabetic nephropathy [6]. The portion of end-stage renal disease (ESRD) attributed to FSGS has increased 11-fold, from 0.2% to 2.3%, over a 21-year period. The annual incidence of membranous nephropathy ESRD also remains high [7]. Thus, timely and accurate diagnosis is extremely important for choosing an adequate therapy and preventing serious consequences.

Urine proteomics can reflect various pathophysiological changes in the body [8], and even more so for renal dysfunction [9]. The search for specific markers among proteins and peptides in the urine is a generally growing trend due to the high information content and good stability of urine as an object of analysis, as well as the absolutely non-invasive method of sample collection, which is extremely important for potential alternatives to the kidney biopsy procedure. Urinary protein or peptide markers have already been proposed for a large number of socially significant pathologies, including a number of cancers and cardiovascular diseases, in addition to disorders associated with renal dysfunction and CKD (as reviewed in [10]).

Mass spectrometric (MS) studies using numerous approaches have made the greatest contribution to obtaining information about changes in the urinary proteome and peptidome in CKD [10,11,12,13,14,15,16]. Nevertheless, of about 100 potential protein markers of different nephropathies described so far, only a few of them have demonstrated consistent reproducibility in at least two different studies: an increase in afamin was observed in membranous nephropathy (MN) [17,18,19]; an increase in antithrombin III was shown in IgA nephropathy (IgAN) [20,21]; an increase in α-1 anti-chymotrypsin was revealed in lupus nephritis (LN) [22,23]; and diabetic nephropathy (DN) was shown to be associated with an increase in zinc-α2-glycoprotein [24,25] and AMBP [25,26], as well as a decrease in transthyretin [23,24]. However, at the same time, changes in the levels of a number of common proteins—such as collagens, uromodulin, serum albumin, and alpha-1-antitrypsin (A1AT) [8]—also remain the focus of much attention as important markers that can have an even greater diagnostic capacity than individual specific markers [11,13,15,16,18,25].

Thus, the development of a proteomic marker panel for distinguishing nephropathies remains highly relevant, so quantitative studies are of particular importance. Multiple-reaction monitoring (MRM) MS technology with internal stable-isotope-labeled standards (SISs) used for normalization has already been successfully applied for the development of quantitative assays with high specificity, precision, and robustness [27,28,29,30]. In particular, a recent targeted MRM-based analysis of 72 plasma samples from patients with CKD confirmed a significant association of the AMBP protein, beta-2-microglobulin (B2M), lysozyme C (LYZ), hemoglobin subunit beta (HBB), and pigment epithelium-derived factor (PEDF) with the estimated glomerular filtration rate (eGFR), and the estimation of absolute plasma concentrations of a combination of these markers provided a stronger association with the outcome of CKD than individual markers [31].

Recently, we applied MRM analysis to confirm the significance of 47 urine protein markers selected from label-free MS analysis for distinguishing FSGS and minimal change disease (MCD) [32]—high-proteinuritic primary podocytopathies with nephrotic syndrome and potential for misclassification [33]. Of the 22 significantly different proteins selected in this study, alpha-2-HS-glycoprotein, complement C9, clusterin, β-2-microglobulin, retinol-binding protein 4, plasminogen, complement C4, prothrombin, vitamin-D-binding protein, hemopexin, transthyretin, and cystatin C were consistent with previously proposed markers for various nephropathies associated with CKD [12,15,16,17,21,24,25,34]. Meanwhile, only the first five of them were previously proposed as potential markers for FSGS or MCD [12,15,17,34].

There have been few studies regarding the development of highly multiplexed MRM assays for urine proteome analysis. Percy et al. developed an MRM panel for quantitative analysis of 136 potential urinary protein biomarkers and successfully applied it for patients with prostate cancer [29]. A targeted urine proteome assay (TUPA) for quantification of 167 urinary proteins in kidney diseases was further proposed by Cantley et al. [30], and nine kidney transplant patients with immediate or delayed graft function were compared. The consistency of both MRM assays is rather low and consists of only 22 common peptides among a total of 415 unique peptides (Percy et al.—213; TUPA—224). Thus, the further development of targeted urine proteome assays for CKD is a pressing task.

In the present study, we considered a panel of 177 proteins and corresponding peptides from the BAK270 MRM assay, which was previously developed for the analysis of potential protein biomarkers in blood plasma [28]. Because proteinuria associated with renal impairment is usually associated with an increased diversity of plasma proteins being present in urine, we considered the use of this panel appropriate. Another advantage of the BAK270 MRM assay is that it includes 35 potential CKD markers described previously [10]. Targeted MRM MS analysis was performed for 46 CKD patients with glomerulopathies, including MN (n = 12), FSGS (n = 26), MCD (n = 8), and healthy controls (n = 23). The results obtained confirm a number of previously proposed CKD markers. Combination of MRM analysis with machine learning for data processing allows for an even greater capacity to generate specific classifiers using proteomic marker panels [35,36]. However, both MRM and machine learning in general are not yet widely used methods in the study of urinary proteomic markers, including CKD markers in particular. Therefore, a preliminary assessment of the effectiveness of using a combination of these approaches in relation to urinary proteomic markers was the rationale for the present study. As a result, a highly accurate classifier was developed that allows for distinguishing between mild and severe glomerulopathies based on the assessment of only three urine proteins.

## 2. Results

### 2.1. Identification of Significantly Different Urine Proteins

In the present study, we considered a panel of 177 proteins and corresponding peptides from the BAK270 MRM assay, which was previously developed for the analysis of potential protein biomarkers in blood plasma [28]. Targeted proteomic analysis was performed for 69 urine samples from 46 CKD patients (see details in Materials and Methods—Section 4.1) and 23 healthy controls. Liquid chromatography-multiple reaction monitoring mass spectrometry (LC-MRM MS) analysis revealed 138 proteins that were found in ≥2/3 of the samples from at least one of the groups, which included 84 core proteins common to all of the groups (Figure 1, Appendix A). Comparison of our 138-protein panel with the 167 proteins (TUPA) proposed by Cantley et al. [30] and the panel of 136 urinary proteins developed by Percy et al. [29] revealed 49 and 69 common proteins, respectively, while 89 and 69 proteins were new in our MRM assay for CKD, respectively (Appendix A).

Pairwise correlation among the core proteins revealed a number of highly correlated groups, which should be taken into account (Figure 1B). A total of 40 core proteins were found to be statistically different, mainly between the control and at least one of the other groups, at an uncorrected *p*-value of <0.01 (Table 1); all of them passed the 10% FDR cutoff after the Benjamini–Hochberg multiple test correction, and 28 of them passed the 5% FWER cutoff after the Bonferroni–Holm correction. It is important to note that these proteins include 17 potential markers of CKD described previously (Table 1); hence, along with the other 23 significantly different core proteins, they can be used as non-specific markers of all considered glomerulopathies. The group of proteins including antithrombin-III (ANT3), plasminogen (PLMN), alpha-1-antitrypsin (A1AT), transthyretin (TTHY), and Ig gamma-1 chain C region (IGHG1) should be especially emphasized, since they revealed the greatest number of differences between groups (Figure 2). At the same time, proteins that did not show any differences (such as actin, apolipoprotein D, etc.) may also be important to monitor, in particular for simplified normalization, instead of normalization to the total protein concentration.

Preliminary hierarchical clustering of individual samples using significantly different core proteins separated 100% of the healthy controls from CKD patients and distinguished particular CKD groups such as mFSGS and sFSGS from one another well (Figure 3A). In general, samples from these three groups were part of the three main clusters that stand out on the heatmap. At the same time, MCD samples were mainly co-localized with mFSGS, while the MN group samples were distributed in a 1:2 ratio between the two FSGS clusters. Dimension reduction methods such as t-SNE reliably separated the control group (Figure 3B) and made it possible to additionally distinguish samples with mild and severe nephropathic manifestations with just seven extracted principal components. In general, even a preliminary analysis easily distinguished the control group from the rest and, in addition, showed some differences between mild and severe manifestations of different nephropathies.

### 2.2. Building of a Binary Classifier for Distinguishing Mild and Severe Nephropathies

Due to the rather small number of samples in each pathological group, a binary classifier was built for distinguishing samples with mild and severe manifestations of nephropathies. First of all, it was necessary to perform feature selection. First, the most significant features were selected based on their *p*-values, Cohen’s d effect sizes, and feature importance determined by the “decision tree” algorithm (Appendix A). To evaluate the latter, the model was trained on the entire dataset, increasing the minimum “tree” depth until reaching an AUC-ROC value of 1.0. The top five proteins from each approach were pooled, resulting in a set of 10 proteins: alpha-2-antiplasmin (A2AP), IGHG1, sex-hormone-binding globulin (SHBG), Ras GTPase-activating protein nGAP (NGAP), vitamin-D-binding protein (VTDB), attractin (ATRN), alpha-1-antitrypsin (A1AT), plasminogen (PLMN), glutathione peroxidase 3 (GPX3), and aortic smooth muscle actin (ACTA). Further estimation of the predictive power of each feature by a default logistic regression classifier using a fivefold cross-validation approach reduced the number of features, and the best results were achieved with PLMN, GPX3, A1AT, and SHBG (Appendix A). It is also noteworthy that the first two were also among the proteins that showed the greatest numbers of differences between the pathological groups (Figure 2).

Binary classifiers built using four machine learning algorithms suggest that the best ROC-AUC metrics can be achieved even with just three proteins: GPX3, PLMN, and A1AT or SHBG (Figure 4). Both of these three-protein sets may be highly effective for distinguishing samples from patients with mild and severe disease states.

Thus, analysis of just several core proteins, in addition to separating glomerulopathies from the control group, can confidently identify the severity of the disease. Nonetheless, accurate diagnosis still requires more specific markers for consideration.

### 2.3. Perspective Potential Protein Markers Specific for CKD Glomerulopathies

If potential protein markers prevalent only in certain CKD groups (which are present in >2/3 of patients in at least one pathological state, but may be totally absent in the others) are also taken into consideration, additional proteins that can contribute to group discrimination can be revealed (Table 2). Eleven proteins can strengthen the separation of the control group. Nine proteins were increased in all glomerulopathies; however, the degree of this increase varied in different groups and was correlated with the severity of the pathology. Twenty proteins are common potential markers for sFSGF and MN and can reliably distinguish MN from mFSGS and MCD, which cannot be achieved as effectively using only the core proteins. Carboxypeptidase N catalytic chain (CBPN) and alpha-2-macroglobulin (A2MG) should be emphasized, since their levels are the highest in MN (Figure 5). They can also to some extent contribute to the distinction between MN and sFSGS, but 12 unique proteins are even more significant for distinguishing the sFSGS group from all others (Table 2). Separation of the MCD and mFSGS groups seems to be a more difficult task. However, these groups also demonstrate some proteomic differences: MCD patients show significantly higher levels of keratins (K1C10 and K1C9) among all of the pathological groups, as well as essentially higher levels of alpha-2-HS-glycoprotein (AHSG) and prothrombin (F2) than in mFSGS, while urine samples from mFSGS patients show higher levels of haptoglobin (HPT) and A2MG.

The proteins listed in Table 2 include 11 previously described markers of CKD: AHSG, HPT, OSTP, F2, A2MG, ADIPO, CO3, CO9, K1C10, FGA, and PEDF. Therefore, the quantitative analysis performed in this study confirms their diagnostic potential. Further quantitative analysis of a significantly expanded number of samples could make it possible to use machine learning methods to create a more accurate classifier for diagnosing specific glomerulopathies.

## 3. Discussion

In the present study, we applied a targeted MRM assay previously validated for 270 blood plasma proteins for the quantitative analysis of urine proteins in glomerulopathies compared to healthy controls. Because proteinuria associated with renal impairment is usually associated with an increased diversity of plasma proteins being present in urine, we considered the use of this kit appropriate. Since the severity of proteinuria varies greatly, absolute quantification—which is usually applied for the analysis of blood samples—would not allow a correct comparison of urine samples. This is why the amount of total protein used for trypsinolysis and subsequent MS analysis was normalized for this study. Therefore, the obtained results reflect the relative characteristic changes in the urine proteomic contents. Moreover, the contents of a number of identified core proteins, which did not show significant differences in levels between groups (such as ACTA, GELS, APOD, DIAC, IGHM, etc.), can also be used for normalization instead of the total protein concentration.

The used MRM assay kit actually includes 35 potential CKD markers described previously. Nineteen of them turned out to be among the core proteins with significantly different levels, mostly distinguishing all of the glomerulopathies from the control group (Table 1). Nevertheless, ANT3, PLMN, and A1AT also showed significant differences between mild and severe disease states. These results greatly coincide with those of multiple other studies, which showed a significant increase in alpha-1-antitrypsin levels in almost all nephropathies associated with CKD (as reviewed in [10]), while ANT3 and PLMN were previously shown to be increased only in IgAN and MN, respectively. At the same time, the present study revealed several other significantly differing core proteins, which had not been described previously as CKD markers. Particularly, PZP, IPSP, and IGHG1 also underlined the differences between particular glomerulopathies; the first two are of particular interest, as they distinguish FSGS and MN, unlike any other core protein.

Interaction analysis between 40 proteins that are prevalent only in certain groups (Table 2) revealed many intersecting pathways (Appendix A). However, some interactions should be especially emphasized. In particular, proteins identified in all pathological groups (but not in the control group) were more involved in blood coagulation and fibrinolysis (A2MG, F2, C1IN), acute phase response (HPT, AHSG, F2), regulation of complement activation and defense response (A2MG, C1IN, F2, CBPN, AHSG), and post-translational protein modification and negative regulation of response to wounding (OSTP, AHSG, A2M, F2, C1IN) (Appendix A). At the same time, proteins that were prevalent only in severe cases (e.g., sFSGS and MN) were most associated with multiple processes of lipid metabolism (APOA4, APOM, APOC1, APOC3, PLTP, CO3, PHLD, ADIPO), including regulation of lipoprotein particle assembly, remodeling, and levels (Figure 6). Moreover, severe cases included proteins associated with complement activation and regulation (CO3, CO5, CO9) (Appendix A). While the proteins prevailing in the control samples did not form a specific physiological cluster, for the most part they were involved in cell activation (HSPB1, PRDX1, PRDX2, TIMP2, CAMP), extracellular matrix organization and skeletal development (TIMP2, PRDX1, SPARC), and immune response (HSPB1, PRDX1, PRDX2, SPARC, CAMP, FCN2) (Figure 6, Appendix A).

Of the previously proposed markers mostly revealed in pathological groups in the present study, increased levels of AHSG and HPT in all glomerulopathies coincide well with the findings of other studies, while the increases in CERU, F2, A2MG, and ADIPO were previously shown mainly for IgAN (as reviewed in [10]), but this group was not considered in the present study. The presence of fibrinogen chains in severe cases also has some overlap with previous studies [11,15], including more recently published data on fibrinogen γ-chain as a potential marker of renal interstitial fibrosis in IgAN [37]. Increased levels of OSTP were found in all glomerulopathies, with no significant differences (Table 2). The same was true for AHSG, HPT, and F2, as well as for several proteins that have not previously been suggested as markers of CKD, including CBPN, KV401, and C1IN (Table 2). Of these, AHSG, CBPN, HPT, KV401, C1IN, and F2 can separate mild and severe diseases or even distinguish between sFSGS and MN.

Of the other previously described CKD urine markers, the decreased presence of CO3 in MCD, decreased presence of collagen in glomerulopathies, and decreased levels of FGA in MN (compared to sFSGS) are in good agreement with previously published data. The increase in PEDF in sFSGS revealed here seems particularly interesting, as it may correlate with its previously described increase in plasma [31]. Moreover, it should be noted that a number of new potential markers that could be used to distinguish between severe glomerulopathies—such as APOM, ITOH2, and LUM—were identified. However, the increased diversity of plasma proteins in the urine itself should correlate with the degree of deterioration of the renal function; therefore, it is hardly appropriate to consider many proteins as specific markers. In general, further studies using more balanced and urine-specific protein panels (including uromodulin, a variety of collagens, and previously proposed CKD markers) on larger collections of urine samples will facilitate the validation of particular CKD markers, as well as the creation of accurate differentiating panels.

## 4. Materials and Methods

### 4.1. Study Population

The study cohort consisted of 69 participants, including patients with confirmed diagnoses of MCD (n = 8), FSGS (n = 26), and MN (n = 12) (Table 3), as well as healthy controls (n = 23). Participants were recruited in the Nephrology Department of Sechenov First Moscow State Medical University (Moscow, Russia). Informed consent was obtained from all participants. Healthy controls (12 men and 11 women, aged 19 to 58 years) had no history of nephropathies, with normal kidney function and no proteinuria. The exclusion criteria for patients with nephropathies were as follows: active urinary infection, diabetes mellitus, obesity, severe arterial hypertension (≥160/≥100 mm Hg), liver disease, rheumatic systemic diseases, and stage 5 CKD ESRD. Obese patients and patients with long-term arterial hypertension were excluded from analysis to avoid secondary FSGS. Any rheumatic disease (e.g., lupus, systemic vasculitis, rheumatoid arthritis) was also an exclusion criterion. The patients’ clinical parameters, including total blood protein, revealed no significant correlation with gender (Appendix A).

The diagnostic groups (FSGS, MN) were determined by biopsy and immunohistochemistry. No patient was diagnosed with cancer. The MN group included primary aPLA2R-positive patients.

Patients with FSGS were additionally subdivided into mild (mFSGS) and severe (sFSGS) subgroups using a special index, taking into account the conservation of the renal function and estimated glomerular filtration rate (eGFR), as well as the severity of proteinuria and steroid resistance of the nephrotic syndrome, as described previously [32]. The special index was calculated as follows: the first score was assigned depending on the level of eGFR, the second depending on the severity of proteinuria, and the third depending on the steroid resistance of the nephrotic syndrome (Table 4). Steroid resistance was defined as the absence of a decrease in proteinuria levels after 16 weeks of prednisolone therapy, or a decrease by less than 50% of the baseline level. The renal function was considered to be “saved” if the eGFR (as determined by the CKD-EPI formula, eGFR CKD-EPI) was above 60 mL/min/1.73 m^2^. If the total score was <3, patients were classified as mFSGS, while scores of ≥3 indicated sFSGS.

### 4.2. Urine Sample Preparation for LC-MS

Ten milliliters of the middle portion of freshly collected morning urine was centrifuged at 3000 rpm for 15 min immediately after collection. The supernatant was aliquoted and stored at −20 °C.

Urine proteins were precipitated with ice-cold acetone as described previously [32]. In brief, 0.1 mL urine aliquots were quickly thawed, mixed with 0.5 mL of ice-cold acetone, and incubated overnight at −20 °C. The precipitate was centrifuged (20,000× *g*, 10 min) and dissolved in 50 µL of 8 M urea (200 mM Tris-HCL, pH 8.5). Urine samples of healthy participants were 10-fold concentrated before acetone precipitation using 3 kDa Amicon^®^ filters (Millipore, Germany). Protein concentration was measured with the BCA assay (Thermo Scientific, Waltham, MA, USA).

Before trypsinolysis, the samples (100 µg of total protein) were reduced with 5 mM dithiothreitol (30 min, +37 °C) and alkylated in the dark with 20 mM iodoacetamide (30 min). TPCK-treated trypsin (Worthington, Franklin, OH, USA) was added at an enzyme:protein ratio of 1:25, and hydrolysis was performed at +37 °C overnight. The reaction was quenched by adding formic acid up to 0.5%. The SIS peptide mixture was spiked in each sample, followed by desalting by solid-phase extraction using plates (Oasis HLB 96-well Microelution Plate, Waters, Taunton, MA, USA). The eluate was lyophilized and dissolved in 0.1% formic acid to a concentration of 0.5 mg/mL for further LC-MS/MS analysis.

The normalization of the amount of total protein was performed before trypsinolysis and subsequent MS analysis, due to the significant variability in the total protein concentrations of the studied urine samples.

### 4.3. Targeted Quantitative LC-MS/MS Using Multiple-Reaction Monitoring (MRM) with Stable-Isotope-Labeled Peptide Standards (SISs)

Targeted quantitative LC-MS analysis was carried out using synthetic stable-isotope-labeled internal standards (SISs) and natural (NAT) synthetic proteotypic peptides for measuring the corresponding proteins in urine. The selected 270 SISs and NAT synthetic peptides had been previously validated for use in LC/MRM-MS experiments for blood plasma [38]. LC-MS parameters, such as the LC gradient and the MRM parameters (Q1 and MRM scans), were adapted and optimized based on previous studies [38]. The SIS peptide mixture was spiked in each urine sample at a balanced concentration, which was optimized in experiments with dilution of a series of urine samples with proteinuria. Standard curves were generated using NAT and SIS peptide standards with a pooled urine sample as a matrix, as previously described in detail for blood plasma analysis [38].

All samples were analyzed in duplicate by HPLC-MS using an ExionLC™ UHPLC system (Thermo Fisher Scientific, USA) coupled online with a SCIEX QTRAP 6500+ triple-quadrupole mass spectrometer (SCIEX, Toronto, ON, Canada). LC-MS parameters, such as the LC gradient and MRM parameters (Q1 and MRM scans), were adapted and optimized based on previous studies [39,40].

The loaded sample volume was 10 μL per injection. HPLC separation was carried out using an Acquity UPLC Peptide BEH column (C18, 300 Å, 1.7 μm, 2.1 mm × 150 mm, 1/pkg) (Waters, USA) with gradient elution. Mobile phase A was 0.1% FA in water; mobile phase B was 0.1% FA in acetonitrile. LC separation was performed at a flow rate of 0.4 mL/min using a 53 min gradient from 2 to 45% of mobile phase B. Mass spectrometric measurements were carried out using the MRM acquisition method. The electrospray ionization (ESI) source settings were as follows: ion spray voltage 4000 V, temperature 450 °C, ion source gas 40 L/min. The corresponding transition list for MRM experiments, with retention time values and Q1/Q3 masses for each peptide, is available in Appendix A.

For quantitative analysis of the raw LC-MS/MS data, Skyline Quantitative Analysis software (version 20.2.0.343, University of Washington) was used [41,42]. To calculate the peptide concentrations in the measured samples, calibration curves were generated using 1/(x × x)-weighted linear regression methods.

### 4.4. Data Analysis

Statistical analysis and data visualization were performed on Python (3.7.3) with the following packages: SciPy [43], Seaborn [44], Matplotlib [45], and Pandas [46]. Significant differences in protein concentrations in the patient groups were estimated using the Mann–Whitney U-test. The false discovery rate (FDR) control Benjamini–Hochberg procedure and the familywise error rate (FWER) approach with the Bonferroni–Holm method were used to control and prevent the false rejection of hypotheses (type I error). Pearson’s correlation coefficient was used to evaluate the correlation between features and build a correlation matrix. To build a Venn diagram (http://bioinformatics.psb.ugent.be/webtools/Venn/ (accessed on 7 April 2003)), all proteins that were present in ≥2/3 of the samples from at least one of the groups were considered. Only proteins identified in ≥70% of samples of any group were considered for binary classification, reducing the dataset from 138 to 58 features (Appendix A). Since missing values often represent low abundant measurements, the “Nan” values were filled with a Gaussian distribution using the Perseus software [47], with parameters of shift down = 0.4 and width = 0.2 of the mean value for each group. Heatmap hierarchical clustering and principal component analysis (PCA) with t-distributed stochastic neighbor embedding (t-SNE) were used for preliminary estimation of the differences between the studied groups and particular samples. Interaction analysis of physiological processes involving potential markers was performed using the STRING method with default parameters, using an available resource (https://string-db.org/ (accessed on 7 April 2003)).

### 4.5. Machine Learning for Binary Classification

All machine learning models were taken from the Scikit-Learn package [48]. All data were Z-scored for normalization. Feature significance ranking was performed using *p*-values, Cohen’s d effect size, and the “decision tree” algorithm (Appendix A). The prediction power of each feature was additionally estimated by a default logistic regression classifier using 5-fold cross validation. The binary classifier was selected from 4 widely used machine learning algorithms, suitable for small datasets: k-nearest neighbors (kNN), logistic regression (LR), random forest (RF) and support-vector machine (SVM) with a linear kernel; a 5-fold cross-validation approach, grid search (Appendix A), and all possible combinations of selected proteins were used to achieve the best performance (Appendix A).

## 5. Conclusions

The obtained results confirm that MRM-MS combined with machine learning is an effective tool for the development of classifiers for the accurate discrimination of different nephropathies related to CKD, as well as for the validation of candidate proteomic markers described previously. As a result, a highly accurate classifier was developed that enables distinction between mild and severe glomerulopathies based on the assessment of only three urine proteins.

Comparison of our 138-protein panel with the 167 proteins (TUPA) proposed by Cantley et al. [30] and the panel of 136 urinary proteins developed by Percy et al. [29] revealed 49 and 69 common proteins, respectively, while 89 and 69 proteins were new in our MRM assay for CKD, respectively. In the present study, 31 proteins previously suggested as potential CKD markers were validated as such.

Quantification of proteins prevalent in only specific pathologies can lead to a more accurate and earlier CKD diagnosis. The development of particularly specific classifiers still requires greater sample collections and further increases in the number of potential CKD markers used for the MRM assay.

## Figures and Tables

**Figure 1 molecules-28-03323-f001:**
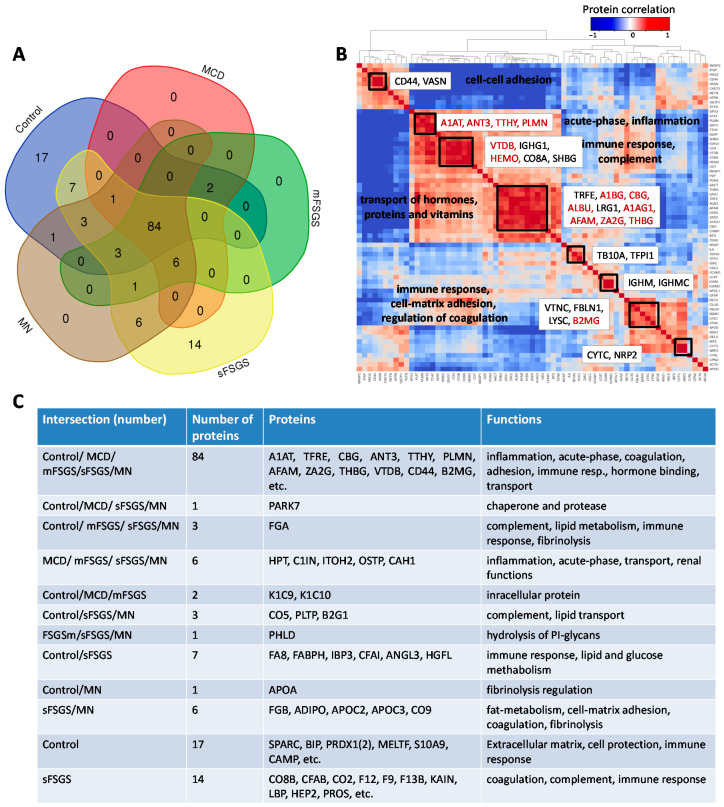
Distribution of 138 proteins from the BAK270 MRM assay that were identified in 69 urine samples from healthy and CKD patients with glomerulopathies: (**A**) Venn diagram for proteins identified in ≥2/3 of samples from at least one group of patients. (**B**) Pairwise correlations among the core proteins. (**C**) Description of proteins in accordance with their distribution in the diagram. Abbreviations for core proteins are given in the legend for Table 1; for other abbreviations, see Appendix A.

**Figure 2 molecules-28-03323-f002:**
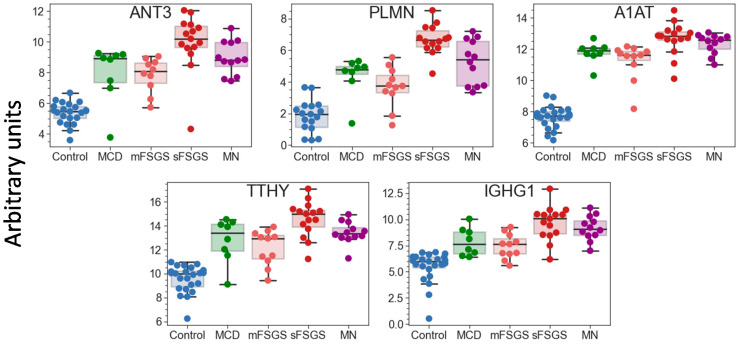
Boxplots for 5 urinary proteins that revealed the greatest number of differences between the control and glomerulopathy groups at FDR < 0.01 (Benjamini–Hochberg). The protein levels are shown in arbitrary units on an Ln(x + 1) scale. Abbreviations: ANT3—antithrombin-III; PLMN—plasminogen; A1AT—alpha-1-antichymotrypsin; TTHY—transthyretin; IGHG1—Ig gamma-1 chain C region.

**Figure 3 molecules-28-03323-f003:**
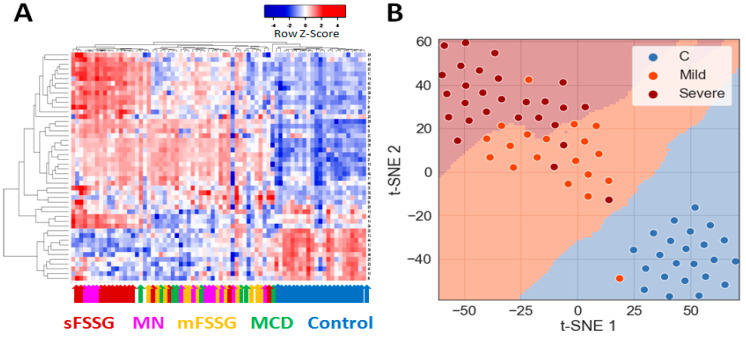
Grouping of all studied urine samples by their levels of 40 core proteins: (**A**) Hierarchical clustering of individual sample data and the 40 significantly different core proteins using the Pearson distance measurement method. The colored arrows indicate the clinical diagnosis of each particular sample. (**B**) Division of samples into three main groups using the PCA and t-SNE methods.

**Figure 4 molecules-28-03323-f004:**
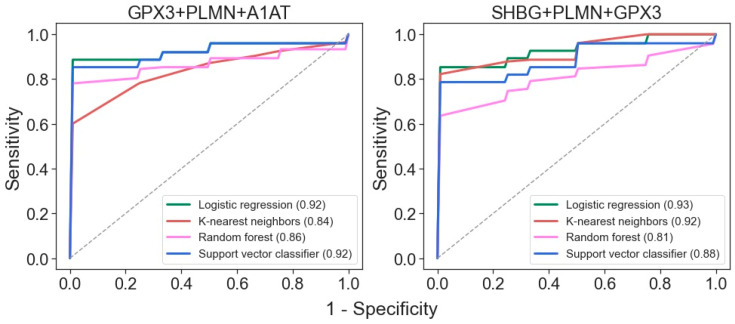
ROC curves for differentiation of glomerulopathies with mild and severe disease states for classifiers generated with the 4 algorithms using the 3-protein sets. AUC values are indicated in brackets.

**Figure 5 molecules-28-03323-f005:**
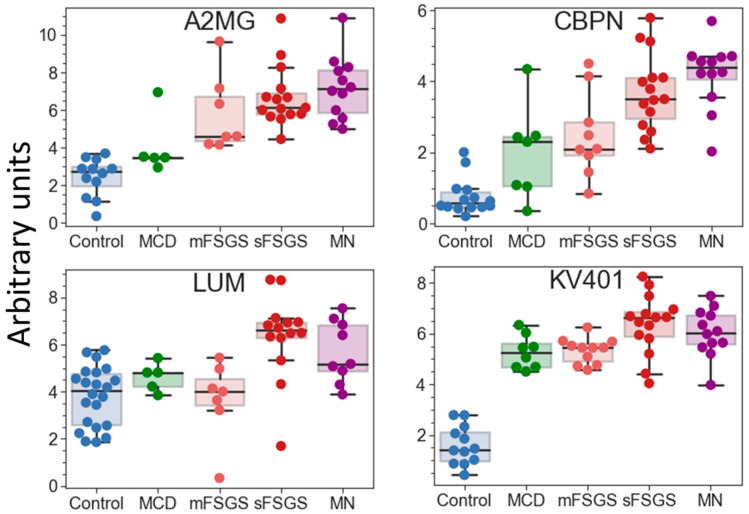
Boxplots for some of the potential protein markers prevalent only in certain glomerulopathies. The protein levels are presented in arbitrary units on an Ln(x + 1) scale. Abbreviations: A2MG—alpha-2-macroglobulin; CBPN—carboxypeptidase N catalytic chain; KV401—immunoglobulin kappa variable 4-1; LUM—lumican.

**Figure 6 molecules-28-03323-f006:**
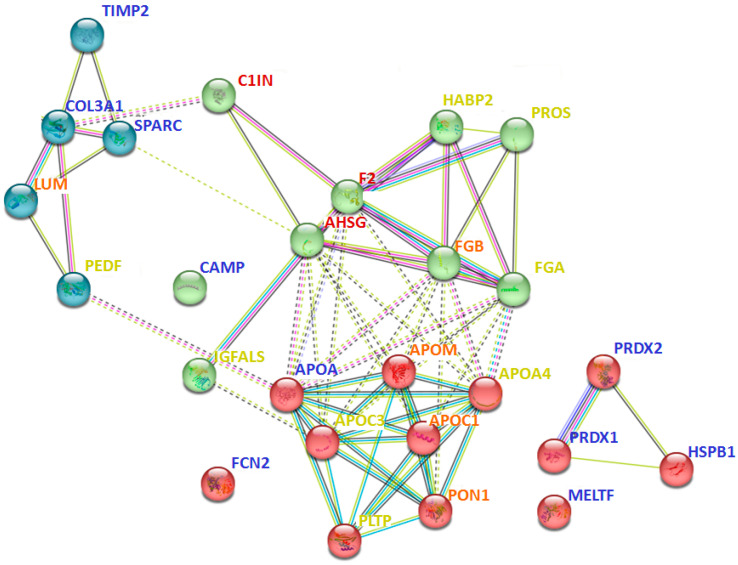
Interaction analysis of some of the potential urinary protein markers prevalent only in certain groups (Table 2), using the STRING method. The red cluster corresponds to different processes associated with lipid metabolism; the green cluster relates to immune processes; the blue cluster corresponds to extracellular matrix organization. The color of the font indicates the predominance of the protein in specific groups: blue—control; yellow—sFSGS; orange—sFSGS and MN; red—MCD, mFSGS, sFSGS, and MN. Abbreviations coincide with the data in Table 2. The more complete information for all prevailing proteins is given in Appendix A.

**Table 1 molecules-28-03323-t001:** The 40 significantly different urinary core proteins identified in all studied groups.

Proteins	Control vs.				sFSGS vs.			mFSGS vs. MN
	MCD	mFSGS	sFSGS	MN	MCD	mFSGS	MN	
ANT3 ^a^, PLMN ^a^	+	+	+	+	+	+	n.d.	n.d.
PZP	+	+	+	+	n.d.	n.d.	+	+
A1AT ^a^, IGHG1	+	+	+	+	n.d.	+	n.d.	n.d.
IPSP	+	+	+	+	n.d.	n.d.	+	n.d.
A1AG1 ^a^, A1BG ^a^, ALBU ^a^, CBG ^a^, CD44 ^a^, LRG1, TRFE, TTHY ^a^, VASN ^a^, ZA2G ^a^, AFAM ^a^, THBG, AACT ^a^, CAD13, PROZ, RETN ^b^	+	+	+	+	n.d.	n.d.	n.d.	n.d.
PON3	+	n.d.	+	+	n.d.	n.d.	n.d.	+
HEMO ^a^	n.d.	n.d.	+	+	+	+	n.d.	n.d.
ATRN ^b^	+	+	+	n.d.	n.d.	n.d.	n.d.	n.d.
BTD ^b^	+	n.d.	+	+	n.d.	n.d.	n.d.	n.d.
GSTP1 ^b^	n.d.	+	+	+	n.d.	n.d.	n.d.	n.d.
KNG1 ^b^	+	+	n.d.	+	n.d.	n.d.	n.d.	n.d.
A2AP	+	n.d.	+	n.d.	n.d.	+	n.d.	n.d.
GPX3	n.d.	n.d.	+	n.d.	+	+	n.d.	n.d.
SHBG	n.d.	n.d.	+	+	n.d.	+	n.d.	n.d.
APOE ^b^	+	+	n.d.	n.d.	n.d.	n.d.	n.d.	n.d.
APOL1, TB10 A ^b^	+	n.d.	n.d.	+	n.d.	n.d.	n.d.	n.d.
VTDB ^a^, CO8A, MASP1, LYAM1 ^b^	n.d.	n.d.	+	+	n.d.	n.d.	n.d.	n.d.
VTNC ^b^	n.d.	+	n.d.	n.d.	n.d.	+	n.d.	n.d.
RET4 ^a,b^	n.d.	n.d.	+	n.d.	n.d.	n.d.	n.d.	n.d.

^a^—Seventeen proteins previously characterized as potential CKD markers; ^b^—ten proteins that did not pass the 5% FWER cutoff after the Bonferroni–Holm correction but passed the 10% FDR cutoff after the Benjamini–Hochberg correction, like all other proteins in the table. The symbol “+” indicates a significant difference after the FDR adjustment (<0.1); “n.d.”—not different. Abbreviations: AACT—alpha-1-antichymotrypsin; A1AG1—alpha-1-acid glycoprotein 1; A1AT—alpha-1-antichymotrypsin; A1BG—alpha-1B-glycoprotein; A2AP—alpha-2-antiplasmin; AFAM—afamin; ALBU—serum albumin; ANT3—antithrombin-III; APOE—apolipoprotein E; APOL1—apolipoprotein L1; ATRN—attractin; BTD—biotinidase; CAD13—cadherin-13; CBG—corticosteroid-binding globulin; CD44—CD44 antigen; CO8A—complement component C8 alpha chain; GPX3—glutathione peroxidase 3; GSTP1—glutathione S-transferase P; HEMO—hemopexin; IGHG1—Ig gamma-1 chain C region; IPSP—plasma serine protease inhibitor; KNG1—kininogen-1; LRG1—leucine-rich alpha-2-glycoprotein; LYAM1—L-selectin; MASP1—mannan-binding lectin serine protease 1; PLMN—plasminogen; PON3—serum paraoxonase/lactonase 3; PROZ—vitamin-K-dependent protein Z; PZP—pregnancy zone protein; RET4—retinol-binding protein 4; RETN—resistin; SHBG—sex-hormone-binding globulin; TB10A—TBC1 domain family member 10A; THBG—thyroxine-binding globulin; TRFE—serotransferrin; TTHY—transthyretin; VASN—vasorin; VTDB—vitamin-D-binding protein; VTNC—vitronectin; ZA2G—zinc-alpha-2-glycoprotein. The significant markers of one or more groups are marked with a gray background.

**Table 2 molecules-28-03323-t002:** Potential protein markers prevalent only in certain groups.

Proteins	Group, % of Samples (Log2 Median)				
	Control	MCD	mFSGS	sFSGS	MN
Alpha-2-HS-glycoprotein (AHSG) ^a^	56.5 (4.86)	100 (8.75)	90.9 (7.48)	100 (11.5)	100 (9.49)
Carboxypeptidase N catalytic chain (CBPN)	60.9 (−0.80)	87.5 (2.45)	81.8 (2.81)	100 (5.01)	100 (6.35)
Haptoglobin (HPT) ^a^	13 (Nan)	62.5 (14.3)	81.8 (15.4)	93.3 (18.5)	91.7 (17.9)
Immunoglobulin kappa variable 4-1 (KV401)	52.3 (−0.92)	100 (7.60)	100 (7.85)	100 (9.58)	100 (8.71)
Osteopontin (OSTP) ^a^	4.34 (Nan)	100 (5.83)	100 (5.73)	100 (6.32)	100 (5.40)
Plasma protease C1 inhibitor (C1IN)	39.1 (Nan)	100 (6.53)	100 (6.59)	86.7 (9.75)	91.7 (7.58)
Prothrombin (F2) ^a^	69.6 (5.90)	87.5 (10.8)	100 (9.28)	93.3 (14.0)	100 (11.5)
Alpha-2-macroglobulin (A2MG) ^a^	52.3 (−1.18)	62.5 (4.62)	63.6 (6.04)	100 (8.85)	100 (10.3)
Adiponectin (ADIPO) ^a^	21.7 (Nan)	25 (Nan)	27.3 (Nan)	73.3 (4.72)	66.7 (3.99)
Apolipoprotein C-I (APOC1)	43.5 (Nan)	25 (Nan)	54.5 (3.80)	80 (10.3)	66.7 (9.60)
Apolipoprotein M (APOM)	82.6 (3.07)	50 (−0.21)	63.6 (0.47)	86.7 (5.53)	66.7 (3.93)
Carbonic anhydrase 1 (CAH1)	13 (Nan)	50 (0.39)	63.6 (1.55)	100 (3.90)	75 (3.60)
Complement C3 (CO3) ^a^	60.9 (2.30)	37.5 (Nan)	63.6 (4.52)	86.7 (13.0)	91.7 (10.3)
Complement component C9 (CO9) ^a^	0 (Nan)	37.5 (Nan)	45.5 (Nan)	80 (10.3)	75 (6.37)
Fibrinogen beta chain (FGB)	13 (Nan)	25 (Nan)	27.3 (Nan)	73.3 (11.6)	75 (9.93)
Inter-alpha-trypsin inhibitor heavy chain H2 (ITOH2)	13 (Nan)	50 (2.86)	63.6 (1.25)	80 (9.14)	75 (7.11)
Lumican (LUM)	95.7 (5.60)	62.5 (5.82)	72.7 (4.58)	86.7 (9.38)	75 (7.20)
Phosphatidylinositol-glycan-specific phospholipase D (PHLD)	0 (Nan)	37.5 (Nan)	45.5 (Nan)	86.7 (3.42)	75 (3.57)
Apolipoprotein(a) (APOA)	100 (7.63)	37.5 (Nan)	45.5 (Nan)	40 (Nan)	58.3 (3.35)
Cathelicidin antimicrobial peptide (CAMP)	100 (4.59)	12.5 (Nan)	18.2 (Nan)	13.3 (Nan)	8.3 (Nan)
Creatine kinase B-type (KCRB)	82.6 (2.74)	0 (Nan)	0 (Nan)	6.67 (Nan)	0 (Nan)
Ficolin-2 (FCN2)	87 (5.46)	12.5 (Nan)	9.1 (Nan)	46.7 (Nan)	16.7 (Nan)
Heat shock protein beta-1 (HSPB1)	78.3 (5.02)	12.5 (Nan)	45.5 (Nan)	13.3 (Nan)	25 (Nan)
Melanotransferrin (MELTF)	82.6 (2.94)	0 (Nan)	0 (Nan)	0 (Nan)	0 (Nan)
Metalloproteinase inhibitor 2 (TIMP2)	91.3 (2.29)	0 (Nan)	9.1 (Nan)	26.7 (Nan)	8.3 (Nan)
Peroxiredoxin-1 (PRDX1)	78.3 (3.48)	0 (Nan)	18.2 (Nan)	6.7 (Nan)	8.3 (Nan)
Peroxiredoxin-2 (PRDX2)	69.6 (4.13)	25 (Nan)	27.3 (Nan)	46.7 (Nan)	33.3 (Nan)
SPARC (SPARC)	73.9 (1.46)	0 (Nan)	0 (Nan)	6.7 (Nan)	8.3 (Nan)
Keratin type I cytoskeletal 10 (K1C10) ^a^	69.6 (8.09)	62.5 (9.08)	63.6 (7.07)	6.7 (Nan)	25 (Nan)
Keratin type I cytoskeletal 9 (K1C9)	65.2 (4.30)	62.5 (6.66)	72.7 (4.21)	26.7 (Nan)	16.7 (Nan)
Apolipoprotein A-IV (APOA4)	60.9 (1.51)	62.5 (4.92)	54.5 (5.22)	86.7 (11.3)	58.3 (6.50)
Apolipoprotein C-III (APOC3)	17.4 (Nan)	37.5 (Nan)	45.5 (Nan)	80 (9.18)	58.3 (7.09)
Cartilage acidic protein 1 (CRAC1)	0 (Nan)	0 (Nan)	9.1 (Nan)	66.7 (1.53)	25 (Nan)
Coagulation factor XII (FA12)	0 (Nan)	0 (Nan)	18.2 (Nan)	66.7 (4.61)	25 (Nan)
Complement C5 (CO5)	56.5 (−3.96)	25 (Nan)	45.5 (Nan)	80 (5.49)	58.3 (1.71)
Fibrinogen alpha chain (FGA) ^a^	69.6 (3.08)	25 (Nan)	54.5 (4.13)	86.7 (8.69)	58.3 (5.82)
Heparin cofactor 2 (HEP2)	0 (Nan)	12.5 (Nan)	18.2 (Nan)	73.3 (7.79)	33.3 (Nan)
Hyaluronan-binding protein 2 (HABP2)	0 (Nan)	12.5 (Nan)	18.2 (Nan)	73.3 (4.83)	33.3 (Nan)
Insulin-like growth-factor-binding protein complex acid labile subunit (IGFALS)	17.4 (Nan)	0 (Nan)	9.1 (Nan)	73.3 (4.08)	33.3 (Nan)
Phospholipid transfer protein (PLTP)	82.6 (−0.78)	12.5 (Nan)	18.2 (Nan)	73.3 (2.14)	50 (−1.34)
Pigment epithelium-derived factor (PEDF) ^a^	8.7 (Nan)	25 (Nan)	18.2 (Nan)	73.3 (10.2)	33.3 (Nan)
Vitamin-K-dependent protein S (PROS)	0 (Nan)	12.5 (Nan)	18.2 (Nan)	73.3 (5.18)	33.3 (Nan)

^a^—Thirteen potential markers related to CKD described previously. The most likely markers of one or more groups are marked with a gray background (taking into account the percentage of representation and the median value). “Nan”—missing values that indicate concentrations that are too low to be quantified or completely absent in the sample.

**Table 3 molecules-28-03323-t003:** Characteristics of the 46 CKD patients.

	MCD	mFSGS	sFSGS	MN
n	8	11	15	12
Age (years)	38 (27; 59)	33.5 (27; 50)	50.5 (39; 61)	52.5 (49; 62)
Sex (%, F)	75.0	50.0	30.8	25.0
Proteinuria (g/24 h)	3.24 (1.84; 3.65)	2.3 (1.28; 3.37)	5.0 (4.0; 8.64)	3.25 (2.66; 4.05)
Nephrotic syndrome (%)	37.5	36.4	93.3	41.7
eGFR CKD-EPI ^a^	93 (70.8; 106)	96.9 (71.3; 115.7)	42.0 (37.0; 62.2)	77.3 (44.9; 87.9)
Steroid-resistance (%)	0	30.0	84.6	N.d.
Failure response to other immunosuppressants (%)	0	0	46.2	16.7
Time to remission, months	1 (0.0; 19.3)	1 (0.26; 5.13)	31 (8.8; 58.5)	8 (1.25; 22.3)
Serum tests:				
Serum albumin (mg/mL)	29.3 (19.7; 35.0)	34.1 (29.1; 37.4)	22.6 (17.1; 26.5)	26.5 (21.2; 33.2)
Creatinine (μM)	80.9 (71.4; 100.0)	68.9 (58.2; 95.6)	148 (122; 163)	93.7 (79.0; 124)
Total cholesterol (mM)	7.89 (7.04; 11.0)	6.86 (6.23; 11.5)	9.69 (8.3; 11.9)	7.91 (7.05; 9.48)
Therapy:				
Corticosteroids (%)	100	70	84.6	41.7
Cyclosporin (%)	62.5	50	69.2	25.0
Cyclophosphamide (%)	50	40	61.5	41.7
Micophenolate mofetil (%)	0	10	30.8	0

^a^ eGFR values are expressed in mL/min/1.73 m^2^ and were estimated using the CKD-EPI formula. Abbreviations: FSGS—focal segmental glomerulosclerosis (mFSGS—mild; sFSGS—severe); MCD—minimal change disease; MN—membranous nephropathy; eGFR—estimated glomerular filtration rate. “N.d.”—not determined. The 0.25 and 0.75 percentiles are indicated in brackets next to the presented median values.

**Table 4 molecules-28-03323-t004:** Calculation of the FSGS severity index.

Parameters	Score
eGFR CKD-EPI, mL/min/1.73 m^2^	
>60	0
45–59	1
35–45	2
<35	3
Proteinuria, g/24 h	
>2	0
2–3	0.5
3–4	1
4–5	1.5
5–6	2
6–7	2.5
>7	3
Steroid resistance	
Absent	0
Present	1

## Data Availability

All experimental results from MRM analysis were uploaded to the PeptideAtlas SRM Experiment Library (PASSEL) and are available via the link: http://www.peptideatlas.org/PASS/PASS04823 (accessed on 8 April 2023).

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
