# Peer review of "Targeted MRM Quantification of Urinary Proteins in Chronic Kidney Disease Caused by Glomerulopathies"

_molecules, 2023, doi:10.3390/molecules28083323_

Round 1

Reviewer 1 Report

In the presented manuscript, the authors described the use of targeted MRM-quantification of urinary proteins in chronic 2 kidney disease. This topic is very interesting and up-to-date - as every research concerning biomarkers. However, in my opinion this paper is not sufficient to be published. I have several suggestions and questions:

1. The aim of the study is not clear. It should be emphasized in the abstract and introduction. 

2. A lot of references are old... Almost half of them are more than 5 years.

3. The study group is really small, even if it is preliminary study. Why only 23 controls were used? Is it possible to obtain significant statistical results? 

4. The conclusions are too general. "Obtained results confirm the high potential of quantitative MRM-MS studies for 414 urine proteomics." - this statement is obvious. There is no longer need to confirm usefullness of the proteomic tools to find biomarkers. there are a lot of studies about it. 

5. What are the function of the described proteins? Why they are changed in the studied disease? Are they connected in some metabolic pathways?

To sum up my remarks: The study has no main goal. Research is presented on very small grup of samples. It is obvious that MRM technique is good for proteomic studies, so why are these findings important? Maybe some external validation using independent group of sample should be performed to confirm that presented proteins might be important?

Author Response

In the presented manuscript, the authors described the use of targeted MRM-quantification of urinary proteins in chronic 2 kidney disease. This topic is very interesting and up-to-date - as every research concerning biomarkers. However, in my opinion this paper is not sufficient to be published. I have several suggestions and questions:

Answer: We appreciate the Reviewer’s comments and positive feedback!  We have done our best to improve the text and address all the comments and criticism. The corresponding revised text is highlighted with blue color.

  1. 1. The aim of the study is not clear. It should be emphasized in the abstract and introduction. 

Answer: Thank you for rising this important issue! We agree that the aim of the study need to be emphasize. As far as we know there are only two studies regarding development of highly multiplexed MRM assay for urine proteome analysis. Percy et al. developed an MRM panel for quantitative analysis of 136 potential urinary protein biomarkers and successfully applied it for patient with prostate cancer [30]. A targeted urine proteome assay (TUPA) for quanti-fication of 167 urinary proteins in kidney diseases was further proposed by Cantley et al. [31] and nine kidney transplant patients with immediate or delayed graft function were compared. The consistency of both MRM assays is rather low and consist of only 22 common peptides among overall 415 unique peptides (Percy et al – 213; TUPA - 224). Thus the further development of targeted urine proteome assay for CKD is an actual tasks.

Moreover, combination of MRM-analysis with machine learning for data processing allows for an even greater capacity to generate specific classifiers using proteomic marker panels [32,33]. However, both MRM and machine learning in general are not yet widely used methods in the study of urinary proteomic markers, including CKD markers in particular. Therefore, a preliminary assessment of the effectiveness of using a combination of these approaches in relation to urinary proteomic markers was the rationale for the current study.

As recommended, we revised the text and tried to specify the aim of the study and added the corresponding text in “Abstract ” and “Introduction” and “Conclusion”:

“…Targeted quantitative urine proteome analysis by mass spectrometry (MS) with multiple reaction monitoring (MRM) is a promising tool for early CKD diagnostics that could replace the invasive biopsy procedure. However, there are only few studies regarding development of highly multiplexed MRM assay for urine proteome analysis and the described so far two MRM assays for urine proteomic demonstrate very low consistency. Thus the further development of targeted urine proteome assay for CKD is an actual tasks.”.

Additionally we discussed the previously developed MRM assay for urine proteome analysis:

“Percy et al. developed an MRM panel for quantitative analysis of 136 potential urinary protein biomarkers and successfully applied it for patient with prostate cancer [30]. A targeted urine proteome assay (TUPA) for quanti-fication of 167 urinary proteins in kidney diseases was further proposed by Cantley et al. [31] and nine kidney transplant patients with immediate or delayed graft function were compared. The consistency of both MRM assays is rather low and consist of only 22 common peptides among overall 415 unique peptides (Percy et al – 213; TUPA - 224). Thus the further development of targeted urine proteome assay for CKD is an actual task.”

Also we try to explain our study design with BAK 270 MRM assay:

“Herein, a BAK270 MRM assay previously validated for blood plasma protein analysis was adapted for urine targeted proteomics. Because proteinuria associated with renal impairment is usually associated with an in-creased diversity of plasma proteins being present in urine, the use of this panel was appropriate. Another ad-vantage of the BAK270 MRM assay is that it includes 35 potential CKD markers described earlier.”

  1. A lot of references are old... Almost half of them are more than 5 years.

Answer: Unfortunately, there are not so many mass-spectrometric studies on urinary proteomics in CKD and they are not published very often. We have cited works that are really relevant to this study, and we would not like to shorten the reference list by removing references older than 5 years, especially two MRM studies (2015). Nevertheless, we tried to satisfy this remark as much as possible, and cited 2 more works (current references [31] and [37]), one of which was published quite recently (BMC Nephrol. 2023, 24, 60, doi: 10.1186/s12882-023-03103-7).

  1. The study group is really small, even if it is preliminary study. Why only 23 controls were used? Is it possible to obtain significant statistical results?

Answer: As mentioned above the aim of the further development of targeted urine proteome assay for CKD. The total number of samples used was generally at the level of many other cited studies (in particular, current ref. [8, 13, 17, 18, 20, 31, 37]) and was quite sufficient to show a striking difference not only between controls and nephropathies, but also between mild and severe pathologies. It is important to take into account that urine as an object of study in this respect differs significantly from blood plasma and is characterized by really noticeable differences in pathological changes. In addition, it should be noted that our task was not to discover new markers, but rather to adapt new protein/peptide panel from BAK 270 and demonstrated its possibilities of CKD study in combination with machine learning. We hope that the specification of the aim of the study (according to your 1st question) also clarifies this issue.

  1. The conclusions are too general. "Obtained results confirm the high potential of quantitative MRM-MS studies for 414 urine proteomics." - this statement is obvious. There is no longer need to confirm usefullness of the proteomic tools to find biomarkers. there are a lot of studies about it.

Answer: As recommended, the text was revised in conclusions and this phrase was removed.

  1. What are the function of the described proteins? Why they are changed in the studied disease? Are they connected in some metabolic pathways?

Answer: According to this comment, we performed additional STRING analysis, added Fig. 6 and a Supplementary Table (with current number S5) with corresponding discussion, and somewhat expanded conclusions.

To sum up my remarks: The study has no main goal. Research is presented on very small grup of samples. It is obvious that MRM technique is good for proteomic studies, so why are these findings important? Maybe some external validation using independent group of sample should be performed to confirm that presented proteins might be important?

Answer: We agree that the aim of the study need to be emphasize. We hope that the specification of the aim of the study (according to your 1st question) also clarifies this issue. As mentioned before there are only few studies regarding development of highly multiplexed MRM assay for urine proteome analysis and the described so far two MRM assays for urine proteomic demonstrate very low consistency. Additional analysis of our MRM data was performed to demonstrate more clearly the actuality of our study. Comparison of our 138 protein panel with 167 proteins (TUPA) proposed by Cantley et al. [31] and 136 urinary protein panel developed by Percy et al [30] revealed 49 and 69 common proteins correspondently and 89 and 69 proteins were new in our MRM assay for CKD. In the current study, 31 proteins earlier suggested as potential CKD markers, were validated.

Moreover, obtained results confirm that MRM-MS combined with machine learning is an effective tool for development of classifiers for accurate discrimination of different nephropathies related to CKD, as well as for validation of candidate proteomic markers described earlier. As a results a highly accurate classifier has been developed (AUC=0.99) that allows distinguishing between mild and severe glomerulopathies based on the assessment of only three urine proteins (GPX3, PLMN, A1AT or SHBG).

The corresponding revised text is highlighted with blue color.

Reviewer 2 Report

Article Title: Targeted MRM-quantification of urinary proteins in chronic kidney disease caused by glomerulopathies.

This manuscript explores and analyse the proteins in urine samples from the control and chronic kidney disease patients. Here authors have attempted to identify the biomarkers of CKD disease by quantifying these proteins using an established MRM based mass spectrometry analysis. with the help of the obtained results they also use machine learning methods to identify 3 core proteins that can distinguish between mild and severe manifestations of CKD patients. Here the experimental design and the number of samples/study subjects used are well thought. The study is very clear from start to end with clear aims and data interpretation. Paper reads very well and I have only minor corrections to report.

Supplementary S6 is not complete. Data missing.

Line 89: please expand MCD. Not sure what it is.

Author Response

This manuscript explores and analyse the proteins in urine samples from the control and chronic kidney disease patients. Here authors have attempted to identify the biomarkers of CKD disease by quantifying these proteins using an established MRM based mass spectrometry analysis. with the help of the obtained results they also use machine learning methods to identify 3 core proteins that can distinguish between mild and severe manifestations of CKD patients. Here the experimental design and the number of samples/study subjects used are well thought. The study is very clear from start to end with clear aims and data interpretation. Paper reads very well and I have only minor corrections to report.

Answer: We appreciate the Reviewer’s comments and positive feedback! We have done our best to improve the text and address all the comments. Also according to recommendation for other referees we revised Abstract, Introduction and Conclusion sections to specify the aim of the study more clearly! The corresponding revised text is highlighted with blue color.

Supplementary S6 is not complete. Data missing.

Answer: We apologize for table format. We tried to revised its format to make it more clear. This table (now Table S8) just contains the used parameters of the algorithms for GridSearch. It may look unusual, but it contains all necessary details.

Line 89: please expand MCD. Not sure what it is.

Answer: We apologize for this error. The abbreviation has now been explained - minimal change disease (MCD).

Reviewer 3 Report

The authors performed proteomic investigations in urine samples obtained from patients with glomerulopathies using machine learning techniques as classificatory tools. The manuscript is well-designed and contains a nice amount of information; however, it will be improved for publication after some aspects to be considered. These issues are listed and discussed in the following topics:

(1) some of the abbreviations (i.e. MCD) were not defined the first time they appear in the manuscript. Please check this aspect in all the manuscript.  

(2) is unclear the original contribution of the present manuscript. The authors must say clearly this aspect in the introduction of the manuscript. Machine learning approaches are being widely used for data analysis in proteomics. Was this approach previously used for glomerulopathies? This information is unclear in the manuscript.

(3) “PCA analysis also reliably separated the control group (Fig. 3B) and made it possible to 170 additionally distinguish samples with mild and severe nephropathic manifestations with 171 just 7 extracted principal components”. I did not identify the plot of PC1 against PC2 and the vector plot in figure 3b. The sentence may be better written.

(4) The physiological role of the most expressive protein changes was poorly discussed. The biochemical mechanisms involved in the protein changes must be described if available.t

(5) There are some expressive differences in gender among the groups. The authors may discuss how this aspect may impact on their results. Table 3 is more suitable for results and not for methodology, as a characterization of study sample.

Author Response

 The authors performed proteomic investigations in urine samples obtained from patients with glomerulopathies using machine learning techniques as classificatory tools. The manuscript is well-designed and contains a nice amount of information; however, it will be improved for publication after some aspects to be considered. These issues are listed and discussed in the following topics:

Answer: We appreciate the Reviewer’s comments and positive feedback!  We have done our best to improve the text and address all the comments and criticism. The corresponding revised text is highlighted with blue color.

(1) some of the abbreviations (i.e. MCD) were not defined the first time they appear in the manuscript. Please check this aspect in all the manuscript.  

Answer: Thank you! We apologize for this error. The abbreviation has now been explained - minimal change disease (MCD). Сhecked throughout the text.

(2) is unclear the original contribution of the present manuscript. The authors must say clearly this aspect in the introduction of the manuscript. Machine learning approaches are being widely used for data analysis in proteomics. Was this approach previously used for glomerulopathies? This information is unclear in the manuscript.

Answer: Thank you for rising this important issue! We agree that the aim of the study need to be emphasize. As far as we know there are only two studies regarding development of highly multiplexed MRM assay for urine proteome analysis. Percy et al. developed an MRM panel for quantitative analysis of 136 potential urinary protein biomarkers and successfully applied it for patient with prostate cancer [30]. A targeted urine proteome assay (TUPA) for quanti-fication of 167 urinary proteins in kidney diseases was further proposed by Cantley et al. [31] and nine kidney transplant patients with immediate or delayed graft function were compared. The consistency of both MRM assays is rather low and consist of only 22 common peptides among overall 415 unique peptides (Percy et al – 213; TUPA - 224). Thus the further development of targeted urine proteome assay for CKD is an actual tasks.

Additional analysis of our MRM data was performed to demonstrate more clearly the actuality of our study. Comparison of our 138 protein panel with 167 proteins (TUPA) proposed by Cantley et al. [31] and 136 urinary protein panel developed by Percy et al [30] revealed 49 and 69 common proteins correspondently and 89 and 69 proteins were new in our MRM assay for CKD.

Moreover, obtained results confirm that MRM-MS combined with machine learning is an effective tool for development of classifiers for accurate discrimination of different nephropathies related to CKD, as well as for validation of candidate proteomic markers described earlier. As a results a highly accurate classifier has been developed (AUC=0.99) that allows distinguishing between mild and severe glomerulopathies based on the assessment of only three urine proteins (GPX3, PLMN, A1AT or SHBG).

The corresponding revised text is highlighted with blue color.

(3) “PCA analysis also reliably separated the control group (Fig. 3B) and made it possible to 170 additionally distinguish samples with mild and severe nephropathic manifestations with 171 just 7 extracted principal components”. I did not identify the plot of PC1 against PC2 and the vector plot in figure 3b. The sentence may be better written.

Answer: The sentence has been rewritten: Dimension reduction methods such as t-SNE reliably separated the control group (Fig. 3B)…”.

(4) The physiological role of the most expressive protein changes was poorly discussed. The biochemical mechanisms involved in the protein changes must be described if available.

Answer: According to this comment, we performed additional STRING analysis, added Fig. 6 and a Supplementary Table (with current number S5) with corresponding discussion, and somewhat expanded conclusions.

(5) There are some expressive differences in gender among the groups. The authors may discuss how this aspect may impact on their results. Table 3 is more suitable for results and not for methodology, as a characterization of study sample.

Answer: Thank you for this important issue! We agree the that there was expressive differences in gender among the groups. On the current stage of the study we didn’t perform such an analysis for MRM urine data. But we performed such a correlation for clinical patient’s parameters with morphology forms including total blood protein and there was no significant correlation with gender (see Fig S1). In our future study such an analysis will be also performed for MRM urine data!

Thank you for the recommendation regarding movement of Table 3 with characterization of studied patient cohort. We agree that in most clinical proteomic studies this data are placed in corresponding Results subsection. Herein we decided not to overload the corresponding Results section as this was the pilot cohort and we tried to focus and discuss more our MRM results regarding main goal of the study - multiplexed MRM assay for urine proteome analysis.

As recommended we added more information and references to the Table 3 in Results section and the corresponding text was added:  “Targeted proteomic analysis was performed for 69 urine samples from 46 CKD patients (see details in Methods - 4.1 section) and 23 healthy controls.”

Round 2

Reviewer 1 Report

Authors made all the important changes. They also proved the importance and novelty of the research. I recommend this manuscript for publication.